# Insomnia, Anxiety, and Depression Symptoms during the COVID-19 Pandemic May Depend on the Pre-Existent Health Status Rather than the Profession

**DOI:** 10.3390/brainsci11081001

**Published:** 2021-07-29

**Authors:** Paweł Wańkowicz, Aleksandra Szylińska, Iwona Rotter

**Affiliations:** Department of Medical Rehabilitation and Clinical Physiotherapy, Pomeranian Medical University in Szczecin, Żolnierska 48, 71-210 Szczecin, Poland; aleksandra.szylinska@gmail.com (A.S.); iwrot@wp.pl (I.R.)

**Keywords:** COVID-19, Hashimoto disease, SLE, GAD-7, PHQ-9, ISI

## Abstract

Despite the high number of studies on mental health among healthcare workers, only a few have attempted to assess the mental health of people with chronic diseases during the COVID-19 crisis. Therefore, the aim of this study is to evaluate the symptoms of insomnia, anxiety, and depression among people with chronic diseases working in healthcare and in other professions. The study participants were divided into two groups. The first group consisted of 441 healthcare workers, and the second consisted of 572 non-healthcare professionals. Correlation analysis showed a strong correlation between autoimmune diseases and an increase in GAD-7 scale, ISI score, and PHQ-9 scale. Therefore, only autoimmune diseases were included for further analyses as a predictor of insomnia, depression, and anxiety. After adjusting the results for gender, age, smoking, dyslipidemia, hypertension, and profession, the group with autoimmune diseases showed a more than a 2-fold increase in the risk of anxiety symptoms, a more than 2.5-fold increase in the risk of depressive symptoms, and a 4-fold increase in the risk of insomnia symptoms. This study shows that, during the COVID-19 pandemic, the incidence of insomnia, anxiety disorders, and depressive disorders may depend on the pre-existent health status of an individual rather than on their profession.

## 1. Introduction

COVID-19 is a contagious disease caused by severe acute respiratory syndrome coronavirus 2 (SARS-CoV-2), which began in December 2019 in Wuhan City, China [1]. The available literature shows that the earliest case of COVID-19 was detected in the second half of November 2019 [2]. COVID-19 spread rapidly to all continents and was subsequently declared a pandemic by the World Health Organization on March 11, 2020. In Poland, the first case of COVID-19 was reported on 4 March 2020. To date, there have been over 135 million confirmed cases and 3 million deaths worldwide. At the time of this analysis, COVID-19 in Poland has killed 57,000 people and infected more than 2.5 million people.

The COVID-19 pandemic not only threatens our physical health but also our mental health [3]. The negative effects of the pandemic on both the social and economic dimensions of a previously well-functioning European system have become evident since its outbreak [4]. The economies of Europe and the rest of the world came to a sudden halt, with many thousands of people losing their jobs. The COVID-19 pandemic majorly altered productivity in many sectors of employment [5]. While some workers were heavily involved in directly countering the spread of the SARS-CoV-2 pandemic, others were forced to cease activity due to lockdowns or job losses, or to completely reorganize their existing activities around remote working. As employees’ mental health can be affected by their workplace, its organization, and work culture, it is quite likely that these stressors, together with the current COVID-19 pandemic, may result in an increase in mental health problems across the globe [6,7]. When faced with a lack of personal protective devices, staying at work longer than previously, the fear of being infected and infecting family, multitasking, and social stigma, those people working in high-risk environments can experience a significant reduction in their well-being. They may even develop psychosomatic symptoms, such as increased sweating, abdominal pain, diarrhea, headaches, dizziness, mood swings, sleep disturbances, and decreased motivation [8]. Moreover, Toyoshima et al. observed that insomnia may be positively related to presenteeism through the mediating role of state anxiety and complaints of cognitive functions [9,10].

Although the COVID-19 pandemic poses a mental health risk to all occupational groups, current research evaluating this association focuses primarily on healthcare workers [11,12]. In Poland, the high mental health burden placed on this group can be related to factors such as the lack of effective COVID-19 treatment, the lack or limitation of personal protective equipment, and inadequate workforce resources, which have all exacerbated previous institutional inefficiencies in the healthcare system. Secondly, medical staff have to bear the ethical burden of deciding whether to withhold admission to hospital wards and whether to withhold resuscitation, as well as making decisions about to whom ventilator therapy should or should not be admitted. Third, they have to take care of patients who are often isolated from their loved ones, providing comfort to dying patients and remotely informing and comforting their family members. Fourth, there is a fear of transmission of coronavirus to family members, which leads to voluntary isolation for many weeks. Fifth, courses and training for medical staff were halted during the pandemic, leading to losses of wages, learning opportunities, and important certifications to function in the profession. Sixth, as a result of cancellations of appointments and treatments, many healthcare workers lost their regular pay. Seventh, medical professionals working in care homes were often locked out of their workplaces for weeks at a time, with little or no proper personal protection, no opportunity for replacement and no support from the system. Eighth, there is a failure to address healthcare workers’ personal health problems on an ongoing basis, such as deteriorating blood pressure, diabetes, and irregular medication intake [13]. These conditions expose healthcare workers to the risk of anxiety, depression, and insomnia. This is supported by a study conducted by our team in which we examined the mental health status of 441 physicians in Poland during the COVID-19 pandemic. Among the entire group of respondents, 64.4% of study participants reported anxiety, 70.7% of participants reported depression, and 58% of participants reported insomnia [14].

Despite the high number of studies on mental health among high-risk groups such as healthcare workers, few studies have attempted to assess the mental health of people with chronic diseases during the COVID-19 crisis. A major cause of disability in society, chronic diseases require frequent visits to outpatient clinics or hospitals and may result in a major deterioration in health status if healthcare is not provided or is limited, which is common in Poland during the COVID-19 pandemic [15]. Current limitations of biochemical, imaging, and functional tests not only jeopardize proper control of staff health conditions but also generate anxiety among them [16]. In addition, the fear of COVID-19 is compounded by the constant rise in conditions that increase the risk of severe COVID-19, as identified by the Centers for Disease Control and Prevention (CDC), including chronic obstructive pulmonary disease (COPD), cancer, heart failure, coronary artery diseases, diabetes, smoking, immunosuppressive conditions, immune deficiencies, and the use of immunosuppressive drugs [17].

Autoimmunological diseases are not the focus of medical attention, especially in Poland. Incidence rates of autoimmune diseases vary from less than 5 per 100,000 for uveitis to over 240 per 100,000 for systemic lupus erythematosus (SLE) and 350 per 100,000 for autoimmune thyroiditis [18,19,20]. However, as the current pandemic shows, these disorders pose a major challenge to healthcare systems worldwide. The causes of autoimmune diseases are largely unknown and thus difficult to identify at a preclinical stage. For this reason, and also due to the fact that vast knowledge in many fields is required to understand them, they are not the most attractive research topic. However, it seems that knowledge of autoimmune diseases may play a key role in addressing COVID-related complications. Symptoms such as fatigue, muscle aches, and brain fog associated with autoimmune diseases often occur in patients with COVID-19 and can persist for a long time. Autoimmune phenomena are enhanced by the presence of antinuclear antibodies in these patients [21]. Bastard et al. observed that autoantibodies against type I interferon are present in patients with severe COVID-19 [22].

To the best of our knowledge, no studies have been conducted on the relation between COVID-19 and insomnia and psychological distress in people with chronic diseases depending on the type of profession. Therefore, the aim of this study is to evaluate the symptoms of insomnia, anxiety, and depression among people with chronic diseases working in healthcare and in other professions.

## 2. Materials and Methods

A hospital-based cross-sectional survey was conducted from 3 May 2020 to 17 May 2020 among individuals in hospitals with clinics or departments in the West Pomeranian region of Poland where patients with COVID-19 were diagnosed or hospitalized. During this period, at the aforementioned hospital facilities and clinics, consecutive subjects who matched the inclusion criteria were asked to complete (in Polish) the 7-item Insomnia Severity Index (ISI; range 0–28 points) [23], the 9-item Patient Health Questionnaire (PHQ-9; range 0–27 points) [24] and the 7-item Generalized Anxiety Disorder Scale (GAD-7; range 0–21 points) [25], followed by an interview with a structured sociodemographic questionnaire. Furthermore, each respondent answered questions regarding basic demographics; any chronic diseases, such as diabetes, heart failure, hypertension, coronary artery disease, COPD, and dyslipidemia; and smoking. Each patient was also asked about the presence of SLE and Hashimoto disease—two of the most common autoimmune diseases in Poland.

The inclusion criteria were as follows: (1) age 18 years or above; (2) diagnosis of chronic diseases (yes or no); (3) type of profession (medical or non-medical); and (4) informed consent before the study. The exclusion criterion was the diagnosis of sleep, anxiety, and depression disorders.

The study participants were divided into two groups. The first group consisted of 441 healthcare workers having direct contact with the patient, and the second group consisted of 572 non-healthcare professionals. Each participant gave informed consent for the study before commencing. Participants included in this study were allowed to discontinue the survey at any time. Full confidentiality of information was ensured. Before starting the study, the Bioethics Committee of the Pomeranian Medical University in Szczecin (KB-0012/26/04/2020/Z) gave consent for the experiment in accordance with the ethical guidelines of the Declaration of Helsinki.

### Statistical Analysis

Statistica v 13.0 software (StatSoft, Tulsa, OK, USA) was used to perform statistical analysis. The distribution of data was tested using the Shapiro–Wilk test. Quantitative data were analyzed using the Mann–Whitney U test. Qualitative data were analyzed based on the *X*^2^ test. If the subgroup size was insufficient, Yates’s correction was used. The relationship between the analyzed parameters was evaluated using univariable and multivariable logistic regression model analysis. The multivariable logistic regression was corrected for potentially distorting data (gender, age, dyslipidemia, diagnosed hypertension, cigarette smoking, and occupation). Differences were statistically significant at *p* ≤ 0.05.

## 3. Results

### 3.1. Comparison of Chronic Diseases and Mean Scores on ISI, GAD-7, and PHQ-9 Scales between Health Professionals and Non-Medical Professions

Healthcare professionals were significantly more likely to smoke cigarettes and more likely to suffer from dyslipidemia than non-medical professionals (*p* < 0.001 and *p* < 0.001, respectively). On the other hand, the non-medical group was significantly more likely to be female, younger, and burdened with autoimmune diseases (such as SLE and Hashimoto disease) compared to the health professional group (*p* < 0.001, *p* < 0.001, *p* < 0.001, respectively). In addition, statistically significant differences in mean GAD-7, PHQ-9, and ISI scores were found between healthcare professionals and non-medical professionals. A comparison of cases is shown in Table 1.

### 3.2. Correlation between the ISI, GAD-7, and PHQ-9 Scales and Selected Parameters

This analysis showed a strong correlation between autoimmune diseases and an increase in GAD-7 scale (r = 0.818, *p* < 0.001), ISI score (r = 0.841, *p* < 0.001), and PHQ-9 scale (r = 0.820, *p* < 0.001) (Table 2). Therefore, only autoimmune diseases were included for further analyses as a predictor of insomnia, depression, and anxiety.

### 3.3. Evaluation of Selected Parameters in Relation to Autoimmune Disease

Analysis of the selected parameters showed statistically significant differences in age (*p* < 0.001), gender (*p* < 0.001), smoking cigarettes (*p* < 0.001), dyslipidemia (*p* = 0.002), hypertension (*p* < 0.001), and profession (*p* < 0.001) between participants with and without autoimmune disease. Moreover, statistically significant differences in mean GAD-7, PHQ-9, and ISI scores were found between participants with and without autoimmune diseases (*p* < 0.001) (Table 3).

### 3.4. Insomnia, Anxiety, and Depression in Patients with Autoimmune Diseases

After the results were corrected for age, gender, diagnosed hypertension, dyslipidemia, cigarette smoking and profession, it was confirmed that the diagnosis of an autoimmune disease was associated with an increased risk of insomnia (ISI, OR = 4.015; *p* <0.001), anxiety (GAD-7, OR = 2.273; *p* < 0.001), and depression (PHQ-9, OR = 2.559; *p* < 0.001) (Table 4).

## 4. Discussion

To the best of our knowledge, this is the first study to examine the effects of the COVID-19 pandemic on insomnia, anxiety, and depression depending on the profession and occurrence of chronic diseases. The biggest finding and strength of this study is that, among the chronic diseases included in this study, Hashimoto disease and systemic lupus erythematosus, two of the most common autoimmune diseases, showed a more than 2-fold increase in the risk of anxiety symptoms, a more than 2.5-fold increase in the risk of depressive symptoms, and a 4-fold increase in the risk of insomnia symptoms.

Current research has only marginally addressed the role of autoimmune diseases as a trigger or exacerbator of mental disorders or insomnia during the COVID-19 pandemic. Two of the current studies were conducted by our team. In the first study, we examined the mental health of a group of 723 respondents divided into two subgroups—those with and without SLE. A significant number of respondents experienced mental health disorders, with a 100% prevalence of anxiety, depression, or insomnia among patients with systemic lupus erythematosus [26]. In study two, we focused on assessing mental health and insomnia among 879 individuals with chronic diseases during the COVID-19 pandemic. Based on correlations among the many chronic diseases included in the analysis, we found a strong association between Hashimoto disease and a higher prevalence of anxiety, depression, or insomnia [27]. Another study, conducted by Ziade et al., used an online survey among 2163 patients with various autoimmune diseases and found a negative impact of the COVID-19 pandemic on mental health in 73% of respondents [28]. Louvardi et al. examined the effect of time spent in quarantine on mental health and somatization among 163 respondents with chronic illnesses, of which 37 were autoimmune patients and 16 healthy respondents. The researchers found that the patients with autoimmune disease showed higher levels of somatization compared to the healthy controls. No significant differences were found between the groups with regard to anxiety or depression [29].

Currently, the published literature indicates that there have been no targeted interventions developed for people with autoimmune diseases during the COVID-19 pandemic. The COVID-19 crisis has mainly been discussed in terms of physical health. However, if the right actions are not taken, this pandemic could in a short time lead to another global crisis in the area of mental health [30]. According to our research, people with autoimmune diseases are especially at risk of developing or increasing anxiety reactions, mood disorders, or sleep disorders, regardless of their place of work. Primarily, this results from the fact that those with autoimmune diseases are at a higher risk of infection, probably due to underlying factors associated with the disease, with comorbidities, or other disorders, or the use of glucocorticosteroids or immunosuppressive drugs [31]. Secondly, uncertainty is a feature of chronic disease, and in the current pandemic in Poland, it is exacerbated by the restriction of sales of drugs modifying the course of systemic lupus erythematosus, such as hydroxychloroquine or chloroquine. Especially during the early stages of the pandemic, this may have generated additional suffering in this group of patients. Third, visits to specialists and biochemical tests, and consequently appropriate health monitoring, have been limited in Poland due to the pandemic. Fourth, autoimmune disease itself is associated with post-traumatic stress symptoms [32], and the media has reported that taking immunosuppressive drugs may lead to a more severe infection of COVID-19 and may additionally negatively affect mental health. Fifth, social isolation or quarantine is a predictor of psychological distress through, for example, decreased access to typical forms of social support or daily physical activity [33]. Finally, due to the aforementioned additional risk of SARS-CoV-2 infection, people with autoimmune diseases may try to distance themselves socially, which may result in a loss of work and consequently a loss of livelihood.

It is also worth noting that autoimmune diseases themselves, even in “normal” conditions, are a source of many stress factors, such as reduced activity, failure to fulfill one’s social or professional roles, and changes in physical appearance, resulting largely from complications of treatment or frequent health problems. These problems are associated, on the one hand, with high costs of healthcare and with financial difficulties on the other [34]. In addition, the low intensity of these stressors in individuals with a diagnosis of autoimmune disease may adversely affect the remission time by exacerbating the symptoms of the disease [35].

Our study also has several limitations. As a cross-sectional study, it did not provide any information on changes in mental health or insomnia over time among the respondents in the study. We were also not able to assess mental health or insomnia before the onset of the COVID-19 pandemic, so it is difficult to draw causal conclusions. The participation in our study was voluntary, and, therefore, individuals who had severe mental problems at the time may not have been available to us and may have been omitted from the study. Finally, although we believe that the results of our survey reflect the immediate response of the selected population to the COVID-19 pandemic outbreak, because the survey was conducted during the first wave of the pandemic in Poland, a place where there has not been a similar epidemiological situation for over 100 years, it is difficult to distinguish the effects of a lockdown alone from the effects of the COVID-19 pandemic.

## 5. Conclusions

This study shows that during the COVID-19 pandemic, the incidence of insomnia, anxiety disorders, and depressive disorders may depend on an individual’s pre-existent health conditions rather than their profession. Individuals with autoimmune diseases, such as systemic lupus erythematosus and Hashimoto disease, are more likely to experience insomnia and psychological distress than patients with other chronic diseases. Therefore, people with systemic lupus erythematosus and Hashimoto disease require special medical, informational, and social support in addition to timely access to necessary medications and tools to reduce stress and provide rest during the current COVID-19 pandemic, as well as after it ends. The results of our study may be helpful for both the government and the medical community in formulating comprehensive interventions to prevent psychological problems among patients with autoimmune diseases.

## Figures and Tables

**Table 1 brainsci-11-01001-t001:** Comparison of selected parameters between health professionals and non-medical professionals.

		Non-Medical Professionals (*n* = 572)	Health Professionals (*n* = 441)	*p*-Value
Gender (*n*, %)	female	418 (73.08%)	229 (51.93%)	<0.001
	male	154 (26.92%)	212 (48.07%)	
Age (years), mean ± SD; Me		38.11 ± 7.29; 37.0	40.23 ± 5.25; 40.0	<0.001
Hypertension (*n*, %)	no	492 (86.01%)	385 (87.30%)	0.551
	yes	80 (13.99%)	56 (12.70%)	
Diabetes mellitus (*n*, %)	no	558 (97.55%)	435 (98.64%)	0.315
	yes	14 (2.45%)	6 (1.36%)	
Coronary heart disease (*n*, %)	no	571 (99.83%)	440 (99.77%)	0.597
	yes	1 (0.17%)	1 (0.23%)	
Heart failure (*n*, %)	no	570 (99.65%)	441 (100.00%)	0.597
	yes	2 (0.35%)	0 (0.00%)	
Dyslipidemia (*n*, %)	no	509 (88.99%)	331 (75.06%)	<0.001
	yes	63 (11.01%)	110 (24.94%)	
Chronic obstructive pulmonary disease (*n*, %)	no	570 (99.65%)	440 (99.77%)	0.821
	yes	2 (0.35%)	1 (0.23%)	
Autoimmune diseases (*n*, %)	no	242 (42.31%)	347 (78.68%)	<0.001
	yes	330 (57.69%)	94 (21.32%)	
Cigarette smoking (*n*, %)	no	515 (90.03%)	221 (50.11%)	<0.001
	yes	57 (9.97%)	220 (49.89%)	
Scale				
GAD-7, mean ± SD; Me		12.88 ± 5.74; 14.5	8.19 ± 5.75; 8.0	<0.001
PHQ-9, mean ± SD; Me		14.96 ± 4.90; 16.0	9.83 ± 6.14;10.0	<0.001
ISI, mean ± SD; Me		17.19 ± 5.59; 19.0	11.83 ± 7.19;14.0	<0.001

Abbreviations: *p*—statistical significance, *n*—number of patients, Me—median, SD—standard deviation, GAD-7—Generalized Anxiety Disorder scale, PHQ-9—Patient Health Questionnaire, ISI—Insomnia Severity Index.

**Table 2 brainsci-11-01001-t002:** Correlation between the ISI, GAD-7, and PHQ-9 scales and selected parameters.

Scale	Selected Parameters	R	*p*-Value
GAD-7	Gender, male	−0.385	<0.001
	Age (years)	−0.080	0.011
	Healthcare professionals	−0.377	<0.001
	Hypertension	−0.098	0.002
	Diabetes mellitus	−0.020	0.517
	Coronary heart disease	0.025	0.423
	Heart failure	−0.027	0.383
	Dyslipidemia	−0.036	0.252
	Chronic obstructive pulmonary disease	−0.019	0.546
	Autoimmune diseases	0.818	<0.001
	Smoke cigarettes	−0.356	<0.001
PHQ-9	Gender, male	−0.370	<0.001
	Age (years)	−0.077	0.014
	Healthcare professionals	−0.408	<0.001
	Hypertension	−0.085	0.007
	Diabetes mellitus	−0.030	0.346
	Coronary heart disease	0.031	0.317
	Heart failure	−0.028	0.377
	Dyslipidemia	−0.041	0.195
	Chronic obstructive pulmonary disease	−0.035	0.268
	Autoimmune diseases	0.820	<0.001
	Smoke cigarettes	−0.352	<0.001
ISI	Gender, male	−0.378	<0.001
	Age (years)	−0.072	0.022
	Healthcare professionals	−0.367	<0.001
	Hypertension	−0.094	0.003
	Diabetes mellitus	−0.030	0.336
	Coronary heart disease	0.025	0.419
	Heart failure	−0.017	0.588
	Dyslipidemia	−0.058	0.064
	Chronic obstructive pulmonary disease	−0.049	0.120
	Autoimmune diseases	0.841	<0.001
	Smoke cigarettes	−0.333	<0.001

Abbreviations: *p*—statistical significance, R—correlation coefficient, GAD-7—Generalized Anxiety Disorder scale, PHQ-9—Patient Health Questionnaire, ISI—Insomnia Severity Index.

**Table 3 brainsci-11-01001-t003:** Evaluation of selected parameters in relation to the occurrence of autoimmune diseases.

		Do You Have Autoimmune Diseases? No (*n* = 589)	Do You Have Autoimmune Diseases? Yes (*n* = 411)	*p*-Value
Gender (*n*, %)	female	275 (46.69%)	372 (87.74%)	<0.001
	male	314 (53.31%)	52 (12.26%)	
Age (years), mean ± SD; Me		39.71 ± 7.07; 39.0	38.11 ± 5.66; 37.5	<0.001
Hypertension (*n*, %)		102 (17.32%)	34 (8.02%)	<0.001
Diabetes mellitus (*n*, %)		15 (2.55%)	5 (1.18%)	0.189
Coronary heart disease (*n*, %)		1 (0.17%)	1 (0.24%)	0.629
Heart failure (*n*, %)		2 (0.34%)	0 (0.00%)	0.629
Dyslipidemia (*n*, %)		119 (20.20%)	54 (12.74%)	0.002
Chronic obstructive pulmonary disease (*n*, %)		3 (0.51%)	0 (0.00%)	0.376
Cigarette smoking (*n*, %)		260 (44.14%)	17 (4.01%)	<0.001
Profession (*n*, %)	Non-medical	242 (41.09%)	330 (77.83%)	<0.001
	Medical	347 (58.91%)	94 (22.17%)	
Scale				
GAD-7, mean ± SD; Me		6.56 ± 4.06; 6.0	16.79 ± 2.81; 17.0	<0.001
PHQ-9, mean ± SD; Me		8.65 ± 4.27; 9.0	18.38 ± 2.57; 18.0	<0.001
ISI, mean ± SD; Me		10.18 ± 4.96; 11.0	21.36 ± 2.40; 21.0	<0.001

Abbreviations: *p*—statistical significance, *n*—number of patients, Me—median, SD—standard deviation, GAD-7—Generalized Anxiety Disorder scale, PHQ-9—Patient Health Questionnaire, ISI—Insomnia Severity Index.

**Table 4 brainsci-11-01001-t004:** Multivariable logistic regression model for autoimmune diseases.

	Autoimmune Diseases			
	*p*-Value	OR	CI − 95%	CI + 95%
**GAD-7**	<0.001	2.273	1.986	2.603
**PHQ-9**	<0.001	2.559	2.196	2.982
**ISI**	<0.001	4.015	2.973	5.424

Abbreviations: *p*—statistical significance, OR—odds ratio, CI—confidence interval, GAD-7—Generalized Anxiety Disorder scale, PHQ-9—Patient Health Questionnaire, ISI—Insomnia Severity Index. Notes: adjusted by gender, age, hypertension, dyslipidemia, cigarette smoking, profession.

## Data Availability

All data that support the findings of this study are available upon request from the corresponding author.

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
