# Peer review of "Insomnia, Anxiety, and Depression Symptoms during the COVID-19 Pandemic May Depend on the Pre-Existent Health Status Rather than the Profession"

_brainsci, 2021, doi:10.3390/brainsci11081001_

Round 1

Reviewer 1 Report

This study suggests that in the COVID-19 pandemic, the incidence of insomnia, anxiety disorders, and depressive disorders may depend on the pre-existent health status, rather than the profession. 

I think that this paper is well written.

To improve the quality of this article, please discuss more about the relationships among insomnia, anxiety, depressive symptoms, and presenteeism in the discussion section. Please refer to the following articles.

Toyoshima, K.; Inoue, T.; Shimura, A.; Uchida, Y.; Masuya, J.; Fujimura, Y.; Higashi, S.; Kusumi, I. Mediating Roles of Cognitive Complaints on Relationships between Insomnia, State Anxiety, and Presenteeism in Japanese Adult Workers. Int. J. Environ. Res. Public Health 202118, 4516. https://doi.org/10.3390/ijerph18094516

Toyoshima K, Inoue T, Shimura A, Masuya J, Ichiki M, Fujimura Y, Kusumi I. Associations between the depressive symptoms, subjective cognitive function, and presenteeism of Japanese adult workers: a cross-sectional survey study. Biopsychosoc Med. 2020 May 4;14:10. doi: 10.1186/s13030-020-00183-x. 

Author Response

Dear Reviewer,

Thank you very much for your valuable comments and advice. We have made changes in accordance with them. Lines 53-55.

Best regards

Reviewer 2 Report

See attachment

Author Response

Dear Reviewer,

Thank you very much for your valuable comments and advice. Below are the answers on your inquiries.

  • Since the assessments were done only once, I don’t think it is correct to say that the presence of an autoimmune correlated with “an increase” in GAD-7, ISI, and PHQ-9 scales. You can either say it was greater at that one time assessment or discuss the additional odds ratio risk as you go on to do. As noted, since there was no baseline comparison the language throughout the manuscript should be very careful to only state at the time of the assessment those with autoimmune diseases were doing much worse.

As the reviewer noted, we first used the correlation analysis between the ISI, GAD-7 & PHQ-9 scales and selected parameters. This analysis showed a strong correlation between autoimmune diseases and an increase in GAD-7 scale (r = 0.818, p <0.001), ISI score (r = 0.841, p <0.001) and PHQ-9 scale (r = 0.820, p < 0.001) (Table 2). Therefore, only autoimmune diseases were included for further analyses as a predictor of insomnia, depression, and anxiety.

Next after the results were corrected for age, gender, diagnosed hypertension, dyslipidemia, cigarette smoking, and profession, it was confirmed that the diagnosis of an autoimmune disease was associated with an increased risk of insomnia (ISI, OR = 4.015; p <0.001 ), anxiety (GAD-7, OR = 2.273; p <0.001) and depression (PHQ-9, OR = 2.559; p <0.001) (Table 4).

  • Page 2: Can you include a specific citation for the development of psychosomatic symptoms mentioned at the end of the first paragraph on page 2?

As suggested by the reviewer, we have include a specific citation. Line 53.

  • “Although the COVID-19 pandemic poses a mental health risk among all occupational groups, current research evaluating this association focuses primarily on health care workers.” The way I am reading this I just don’t think this is true. There are a LOT of studies at this point that have examined the impact of COVID19 on the mental health of the general public, not just health care workers (I’ve provided a few citations that I know of below). If you mean examined mental health risk on specific other occupation groups, like hospitality workers or retail workers, then I think that can be made more clear (though I’m not sure that’s super relevant to your paper). But mental health of the general public has been fairly well documented and at least a few should be very briefly cited and/or discussed here (I don’t know of any in Poland aside from your previous publication, but several used international samples, or nearby country populations):
    • Cunningham, T. J., Fields, E. C., Garcia, S. M., & Kensinger, E. A. (2021). The relation between age and experienced stress, worry, affect, and depression during the spring 2020 phase of the COVID-19 pandemic in the United States. Emotion.
    • Wilson, J. M., Lee, J., & Shook, N. J. (2021). COVID-19 worries and mental health: The moderating effect of age. Aging & Mental Health, 25(7), 1289-1296. https://doi.org/10.1080/13607863.2020.1856778
    • Vahia, I. V., Jeste, D. V., & Reynolds, C. F. (2020). Older adults and the mental health effects of COVID-19. Journal of the American Medical Association, 324(22), 2253-2254. https://doi.org/10.1001/jama.2020.21753
    • Varma, P., Junge, M., Meaklim, H., & Jackson, M. L. (2021). Younger people are more vulnerable to stress, anxiety and depression during COVID-19 pandemic: A global cross-sectional survey. Progress in Neuro-Psychopharmacology and Biological Psychiatry, 109, 110236. https://doi.org/10.1016/j.pnpbp.2020.110236
    • Klaiber, P., Wen, J. H., DeLongis, A., & Sin, N. L. (2021). The ups and downs of daily life during COVID-19: Age differences in affect, stress, and positive events. The Journals of Gerontology: Series B, 76(2), e30-e37. https://doi.org/10.1093/geronb/gbaa096
    • González-Sanguino, C., Ausín, B., Castellanos, M. A., Saiz, J., López-Gómez, A., Ugidos, C., & Muñoz, M. (2020). Mental health consequences during the initial stage of the 2020 Coronavirus pandemic (COVID-19) in Spain. Brain Behavior and Immunity, 87, 172-176. https://doi.org/10.1016/j.bbi.2020.05.040
    • Rodriguez-Seijas, C., Fields, E. C., Bottary, R., Kark, S. M., Goldstein, M. R., Kensinger, E. A., & Cunningham, T. J. (2020). Comparing the impact of CoViD-19-related social distancing on mood and psychiatric indicators in sexual and gender minority (SGM) and non-SGM individuals. Frontiers in psychiatry, 11, 1448.

We will try to explain it in this way, we did not focus on the general public but on specific professions. According to our knowledge, the most research currently concerns health care workers, which we described in the introduction. We agree with the reviewer that there are a lot of studies at this point that have examined the impact of COVID19 on the mental health of the general public

  • As a quick additional note, the fact that the autoimmune effect still came out in your analyses after controlling for gender and age is huge and could be played up, as a common thread in many of the papers I listed above is that older adults had better mental health outcomes during COVID-19, even in spite of the increased risk. Assuming older adults are likely at higher risk for chronic diseases, I think this is telling that it was still enough to put this group in a much worse state of mental health

Probably yes, but it requires careful analysis

  • In the second paragraph on page 2, I think citing even some news articles of the events if possible would benefit the long-term reading of this paper (e.g., lack of resources, withholding admissions, voluntary isolation, loss of pay etc.). And I think there should be a proper citation for your 8th point, failure to address usual health problems.

As suggested by the reviewer, we have include a specific citation. Line  76.

  • I do completely agree that this paper fills a gap by assessing mental health of individuals with chronic disease. I haven’t seen much if at all about this so am enthusiastic about this effort and these results!

We thank the reviewer very much for this observation

  • What were the exact dates of data collection? I ask because some studies that did repeated assessments that I mentioned above (Rodriguez-Seijias et al, 2020; Cunningham et al., 2021; maybe Varma too, I can’t remember) showed that from pretty much March – June mental health metrics largely improved as people coped. I think these studies had international data but were largely US so timeline isn’t guaranteed to be the same, but the point is the first 3 weeks of May may not have even been the worst of it. (see comment in Discussion)

As suggested by the reviewer, we have completed the dates. Lines 115-116.

  • Is there any data on specific lockdown dates in Poland when restrictions may have been worse to help with the context?

Unfortunately, only an epidemic status has been implemented in Poland

  • Was there any qualification for the type of health care professional or the unit they were working on? Would hospital administrators or people not directly taking care of COVID patients be listed under health professionals? If so this should be mentioned as a limitation.

As suggested by the reviewer, we have completed the information. Lines 133.

  • The fact that there was no pre-COVID baseline is obviously a major limitation that will need to be made very clear throughout the results and discussion. All we can say from this data is how these groups in Poland were doing in this specific snapshot in time.

The exclusion criterion in our study was the diagnosis of sleep, anxiety and depression disorders.

  • I am by no means a statistical expert at all on this, but most of the similar studies I’ve seen in this type of work have done linear mixed models and report the β coefficients and confidence intervals. Can the authors briefly explain for me the use of multivariable logistic regression?

Naturally, linear regression is applied to continuous (quantitative) data in our analyzes we focused on the assessment of the presence or not of an autoimmune disease, variables of this type are on a nominal scale. Since the linear regression model is not applicable here, it is necessary to use nonlinear models. The regression model used for nominal variables is the logistic regression model. Therefore, we used a logistic regression. For logistic regression, the β coefficient is not calculated, but logistic regression is based on a specific method of expressing probability, called Odds ratio.

  • Were any corrections made or needed for multiple comparisons with this method?

In our study, we showed differences in data (gender, age, hypertension, dyslipidemia, cigarette smoking, profession), therefore we adjusted our results for confounding variables

  • If only “Man” and “Woman” were options on the demographics, it would be better to report it as “sex” or “biological sex”

We only use the nomenclature indicated by the publisher

  • In Table 1 and 3 could you also include the U or X2 values for reference in the table?

This is not a problem for us, but in our opinion this will extend the table by what not become clear for the reader

  • I think bolding or putting a star next to significant p-values in both tables would be helpful for quickly identifying the significant differences

As suggested by the reviewer, we have made a correction

  • If the primary effects or correlations can be plotted at all I believe it would help with the presentation of the results.

We considered the diagrams for correlation, unfortunately due to the multitude of variables it would be unreadable

  • Again, I just want to reiterate that I think you can really play up that this effect was above and beyond any age effects since you controlled for them.

Age effect was eliminated during the adjustment. Lines 191-192 and Tab.4

  • Again given the timing of the data collection (early May) and the fact that assessments were only conducted once, I think there are serious limitations into how much you can interpret from these results. I DO think they are still worth reporting, but since there are no baseline assessments, do people with autoimmune disease typically have worse mental health and it just persisted during COVID-19? Also there are several different ways that the autoimmune group could have ended up with those higher odds ratios during the assessment period: (1) They could have been worse from the start and stayed worse, (2) everyone could have been really bad early on and the autoimmune people just didn’t recover as fast, (3) autoimmune people might have just been slowly worsening over time, etc. I do not think this is a fatal flaw and I do think these results are interesting and worth reporting, it just needs to be done in a way that recognizes the limitations of how we can interpret them. Again, COVID-19 was a long-term, evolving event and this was just a snapshot

As we mentioned in the exclusion criterion methods was the diagnosis of sleep, anxiety and depression disorders.

  • Page 7: Are those with an autoimmune disease at higher risk of a COVID-19 infection or at higher risk of more adverse outcomes if infected? I know the latter is true, I just genuinely don’t know if they are more prone to be infected at all.

Naturally, these two statements are true. Suffering from an autoimmune disease can generate enormous stress, through a significant reduction in activity at home or at work, financial difficulties related to the cost of medical care and reduced income, lack of acceptance of its appearance resulting from, among other things, complications of the applied treatment, impaired interpersonal relations or loss of independence. Even discreet intensification of everyday stress factors in people with autoimmune diseases affects the hypothalamic-pituitary-adrenal axis homeostasis, which leads to intensification of the disease symptoms or adversely affects the time of its remission.

  • “…people with autoimmune diseases try to distance themselves socially…” I would say “may try to distance themselves socially” unless you have a citation

As suggested by the reviewer, we have made a correction. Lines 240-241.

The authors didn’t really touch on the fact that non-medical professionals actually had much worse mental health reports than health professionals in the discussion. Why do you think that might be? Could it be that medical professionals tend to be highly functioning people and may have adapted by this time better than the general public? Or maybe more hesitancy in reporting their actual level of distress?

We didn’t discuss this topic exactly in the discussion section because in the following tables you can clearly see that, first of all, the status of the profession poorly correlates with the GAD-7, PHQ-9, ISI scales. Second of all, correlation analysis showed a strong correlation between autoimmune diseases and an increase in GAD-7 scale (r = 0.818, p <0.001), ISI score (r = 0.841, p <0.001) and PHQ-9 scale (r = 0.820, p <0.001) (Table 2). Therefore, only autoimmune diseases were included for further analyses as a predictor of insomnia, depression, and anxiety. Our study has shown that in the COVID-19 pandemic, the incidence of insomnia, anxiety disorders, and depressive disorders may depend on the pre-existent health status, rather than the profession. Individuals with autoimmune diseases, such as systemic lupus erythematosus and Hashimoto disease, are more likely to experience insomnia and psychological distress than patients with other chronic diseases

Best regards

Reviewer 3 Report

This is a very interesting and nicely written manuscript.

Author Response

Dear Reviewer,

Thank you for your review.

Best regards